# *DeCUR*: Decoupling Common & Unique Representations for Multimodal Self-supervision

## Abstract

The increasing availability of multi-sensor data sparks interest in multimodal self-supervised learning. However, most existing approaches learn only common representations across modalities while ignoring intra-modal training and modality-unique representations. We propose **De**coupling **C**ommon and **U**nique **R**epresentations (DeCUR), a simple yet effective method for multimodal self-supervised learning. By distinguishing inter- and intra-modal embeddings, De-CUR is trained to integrate complementary information across different modalities. We evaluate DeCUR in three common multimodal scenarios (radar-optical, RGB-elevation, and RGB-depth), and demonstrate its consistent benefits on scene classification and semantic segmentation downstream tasks. Notably, we get straightforward improvements by transferring our pretrained backbones to state-of-the-art supervised multimodal methods without any hyperparameter tuning. Furthermore, we conduct a comprehensive explainability analysis to shed light on the interpretation of common and unique features in our multimodal approach.

## 1 Introduction

Self-supervised learning has achieved break-throughs in machine learning (Ericsson et al., 2022) and many other communities (Krishnan et al., 2022; Wang et al., 2022a). Driven by the success in single modality representation learning, as well as the great potential that large-scale multi-sensor data bears, multimodal self-supervised learning is gaining increasing attention. While image, language and audio (Deldari et al., 2022) have been widely studied, multimodality in other real-world scenarios is lagging behind, such as RGBD indoor scene understanding and multi-sensor Earth observation. In this work, we dig into these important modalities and propose DeCUR, a simple yet effective self-supervised method for multimodal representation learning. We demonstrate the effectiveness of DeCUR on three common multimodal scenarios: Synthetic Aperture Radar (SAR) – multispectral optical, RGB – Digital Elevation Model (DEM), and RGB – depth.

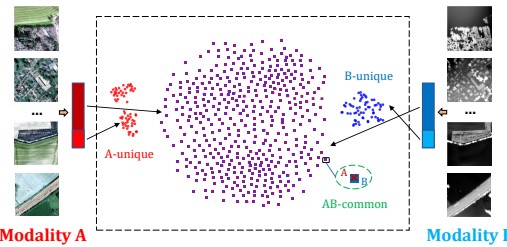

Figure 1: Decoupled common and unique representations across two modalities visualized by t-SNE (Van der Maaten & Hinton, 2008). Each embedding dimension is one data point. red and blue indicate features from modality A and B; red cross and blue square marks indicate (overlapped) common dimensions. Best view in color & zoomed in.

A common strategy for exisiting multimodal self-supervised learning is to use different modalities as augmented views and conduct cross-modal contrastive learning. Such methods follow a similar design of SimCLR (Chen et al., 2020a) and have been widely studied in image-language pretraining. One famous example is CLIP (Radford et al., 2021), where a contrastive loss is optimized for a batch of image-text pairs. However, these methods have common disadvantages such as requiring negative samples and a large batch size, which limit the performance on smaller-scale but scene-complex datasets. To tackle these issues, we revisit Barlow Twins (Zbontar et al., 2021), a redundancy reduction based self-supervised learning algorithm that can work with small batch size, and does not rely on negative samples. Barlow Twins works by driving the normalized cross-correlation matrix

of the embeddings of two augmented views towards the identity. We show that Barlow Twins can be naturally extended to multimodal pretraining with modality-specific encoders, and present its advantages over exsiting methods with contrastive negative sampling.

More importantly, most existing multimodal studies focus only on common representations across modalities (Scheibenreif et al., 2022; Radford et al., 2021; Wang et al., 2021; Girdhar et al., 2023), while ignoring intra-modal and modality-unique representations. This forces the model to put potentially orthogonal representations into a common embedding space, limiting the model's capacity to better understand the different modalities. To solve this problem, we introduce the idea of decoupling common and unique representations. This can be achieved by as simple as separating the corresponding embedding dimensions. During training, we maximize the similarity between common dimensions and decorrelate the unique dimensions across modalities. We also introduce intra-modal training on all dimensions, which ensures the meaningfulness of modality-unique dimensions, and enhances the model's ability to learn intra-modal knowledge.

In addition, little research has been conducted on the explainability of multimodal self-supervised learning. While multiple sensors serve as rich and sometimes unique information sources, existing works like Gur et al. (2021) only consider a single modality. To bridge this gap, we perform an extensive explainability analysis on our method. We visualize the saliency maps of common and unique representations and analyse the statistics from both spatial and spectral domain. The results provide valuable insights towards the interpretation of multimodal self-supervised learning.

## 2 RELATED WORK

**Self-supervised learning**   Self-supervised learning with a single modality has been widely studied. Following the literature, it can be categorized into three main types: generative methods (e.g. Autoencoder (Vincent et al., 2010) and MAE (He et al., 2022)), predictive methods (e.g. predicting rotation angles (Gidaris et al., 2018)) and contrastive methods (joint embedding architectures with or without negative samples). Contrastive methods can be further categorized into four strategies of self-supervision: 1) contrastive learning with negative samples (e.g. CPC (Oord et al., 2018), SimCLR (Chen et al., 2020a) and MoCo (He et al., 2020)); 2) clustering feature embeddings (e.g. SwAV (Caron et al., 2020)); 3) knowledge distillation (e.g. BYOL (Grill et al., 2020), SimSiam (Chen & He, 2021) and DINO (Caron et al., 2021)); 4) redundancy reduction (e.g. Barlow Twins (Zbontar et al., 2021) and VICReg (Bardes et al., 2021)). While most exisitng multimodal works are closely related to the first strategy, DeCUR belongs to redundancy reduction as a natural extension of Barlow Twins that does not require negative samples. DeCUR's decoupling strategy can be perfectly integrated into a simple correlation-matrix-based loss design in Barlow Twins (in VICReg it is also possible to apply but introduces complexity and more hyparameters).

**Multimodal self-supervised learning**   The idea of contrastive self-supervised learning can be naturally transferred to multimodal scenarios, as different modalities are naively the augmented views for the joint embedding architectures. Currently, contrastive learning with negative samples has been mostly developed: CLIP (Radford et al., 2021) for language-image, VATT (Akbari et al., 2021) for video-audio-text, Scheibenreif et al. (2022) for radar-optical, and IMAGEBIND (Girdhar et al., 2023) for a joint embedding of six different modalities. Different from these methods, we propose to explore the potential of negative-free methods by extending the redundancy reduction loss of Barlow Twins. On the other hand, we share an insight with Yang et al. (2022) and Wang et al. (2022b) that intra-modal representations are important complements to cross-modal representations. In addition, we take one step further to decouple common and unique information from different modalities.

**Modality decoupling**   While not widely explored in multimodal self-supervised learning, modality decoupling has been proved beneficial in supervised learning. Xiong et al. (2020; 2021) studied multimodal fusion from network architecture, proposing modality separation networks for RGB-D scene recognition. Peng et al. (2022) investigated modality dominance from the angle of optimization flow, proposing on-the-fly gradient modulation to balance and control the optimization of each modality in audio-visual learning. Zhou et al. (2023) observed feature redundancy for different supervision tasks, proposing to decompose task-specific and task-shared features for multitask learning in recommendation system. Different from the above, we directly perform modality decoupling on the embeddings by separating common and unique dimensions. This simple strategy neither re-

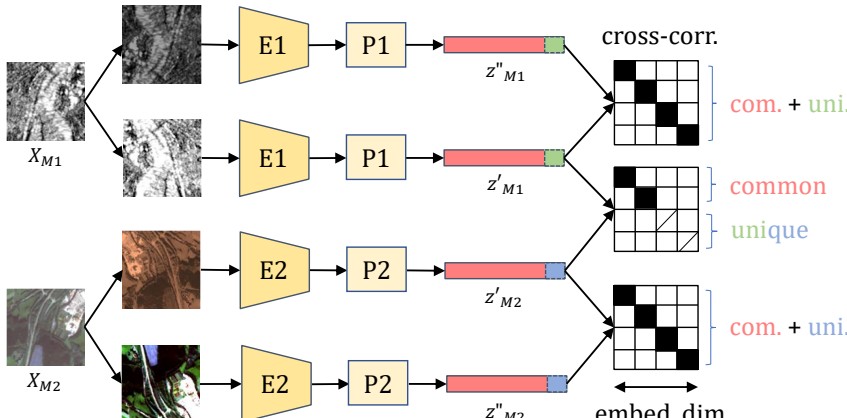

Figure 2: The general structure of DeCUR. $M1$ and $M2$ represent two modalities. Black and white color in the cross-correlation matrices represent 1 and 0 respectively. Two augmented views from each modality are fed to modality-specific encoders ($E1$, $E2$) and projectors ($P1$, $P2$) to get the embeddings $Z$. For cross-modal embeddings, the dimensions are separated into common and unique ones. The correlation matrix of the common dimensions is optimized to be close to the identity, while that of the unique ones to zero. For intra-modal embeddings, both common and unique dimensions are used for the correlation matrix which is optimized to be close to the identity. This naturally helps maintain the meaningfulness of the unique dimensions. In total, DeCUR decouples modality-unique embeddings and learns both intra- and inter-modal representations.

quires architecture modification nor supervision guidance, thus fitting well the generalizability and transferability of self-supervised learning.

## 3 METHODOLOGY

Our main contribution lies in a simple loss design to decouple meaningful modality-unique representations that enhances unsupervised pretraining. Figure 2 presents the general structure of DeCUR. As a multimodal extension of Barlow Twins, DeCUR performs self-supervised learning by redundancy reduction in the joint embedding space of augmented views from intra-/inter-modalities.

Given a batch of multimodal input pairs $X_{M1}$ and $X_{M2}$, two batches of augmented views $X_{M1}'$ and $X_{M1}''$ (or $X_{M2}'$ and $X_{M2}''$) are generated from each modality. Each of the four batches is then fed to a modality-specific encoder and projector, producing batches of embeddings $Z_{M1}'$, $Z_{M1}''$, $Z_{M2}'$ and $Z_{M2}''$ respectively. Batch normalization is applied on each batch of embeddings such that they are mean-centered along the batch dimension. Next, multimodal redundancy reduction is performed on the cross-correlation matrices $\mathcal{C}$ of the embedding vectors.

$$
\mathcal{C}_{ij} = \frac{\sum_b z_{b,i}^A z_{b,j}^B}{\sqrt{\sum_b \left(z_{b,i}^A\right)^2} \sqrt{\sum_b \left(z_{b,j}^B\right)^2}}
\tag{1}
$$

where $Z^A$, $Z^B$ are two embedding vectors, $b$ indexes batch samples, and $i$, $j$ index the dimension of the embedding vectors. $\mathcal{C}$ is a square matrix with size the dimensionality of the embedding vectors, and with values comprised between -1 and 1.

### 3.1 CROSS-MODAL REPRESENTATION DECOUPLING

While most multimodal self-supervised learning algorithms consider only common representations, we introduce the existence of modality-unique representations and decouple them during training. This can be naively done by separating embedding dimensions $K_c$ and $K_u$ to store common and unique representations respectively. The common representations should be identical across modalities, while the modality-specific unique representations should be decorrelated.

On the one hand, a sub-matrix $\mathcal{C}_c$ with size $K_c \times K_c$ is generated from only the common dimensions of the embedding vectors $Z_{M1}{}'$ and $Z_{M2}{}'$ for both modalities. The redundancy reduction loss for the cross-modal common representations reads:

$$\mathcal{L}_{common} = \sum_i \left(1 - \mathcal{C}_{\mathrm{c}ii}\right)^2 + \lambda_c \cdot \sum_i \sum_{j \neq i} \mathcal{C}_{\mathrm{c}ij}^2 \tag{2}$$

where $\lambda_c$ is a positive constant trading off the importance of the first invariance term (to make the common embeddings invariant to the input modalities) and the second redundancy reduction term (to decorrelate the embedding vector components and avoid model collapse).

On the other hand, a sub-matrix $\mathcal{C}_u$ with size $K_u \times K_u$ is generated from only the unique dimensions of the embedding vectors $Z_{M1}{}'$ and $Z_{M2}{}'$ for both modalities. The redundancy reduction loss for the cross-modal unique representations reads:

$$\mathcal{L}_{unique} = \sum_i \mathcal{C}_{\mathrm{u}ii}^2 + \lambda_u \cdot \sum_i \sum_{j \neq i} \mathcal{C}_{\mathrm{u}ij}^2 \tag{3}$$

where $\lambda_u$ is a positive constant trading-off the importance of the first decorrelation term (to decorrelate different modalities) and the second redundancy reduction term (to decorrelate the embedding vector components). However, pure decoupling doesn't ensure the meaningfulness of the unique dimensions, i.e., they could collapse into random decorrelated values. To tackle this issue, we further introduce intra-modal representation enhancement that covers both common and unique dimensions within each modality.

## 3.2 Intra-modal representation enhancing

To ensure the meaningfulness of the unique representations (i.e., avoid collapse of being randomly decorrelated), as well as to enhance intra-modal representations, we introduce intra-modal training that covers both common and unique dimensions. For each modality, a cross-correlation matrix $\mathcal{C}_{\mathrm{M1}}$ (or $\mathcal{C}_{\mathrm{M2}}$) is generated from the full dimensions of the embedding vectors $Z_{M1}{}'$ and $Z_{M1}{}''$ (or $Z_{M2}{}'$ and $Z_{M2}{}''$). The redundancy reduction losses for the intra-modal representations reads:

$$\mathcal{L}_{M1} = \sum_i \left(1 - \mathcal{C}_{\mathrm{M1}ii}\right)^2 + \lambda_{M1} \cdot \sum_i \sum_{j \neq i} \mathcal{C}_{\mathrm{M1}ij}^2 \tag{4}$$

$$\mathcal{L}_{M2} = \sum_i \left(1 - \mathcal{C}_{\mathrm{M2}ii}\right)^2 + \lambda_{M2} \cdot \sum_i \sum_{j \neq i} \mathcal{C}_{\mathrm{M2}ij}^2 \tag{5}$$

where $\lambda_{M1}$ and $\lambda_{M2}$ are positive constants trading off the importance of the invariance term and the redundancy reduction term.

Combining the cross-modal common and unique and intra-modal loss terms, the overall training objective of DeCUR reads:

$$\mathcal{L} = \mathcal{L}_{common} + \mathcal{L}_{unique} + \mathcal{L}_{M1} + \mathcal{L}_{M2} \tag{6}$$

## 4 Implementation details

**Pretraining datasets**  We pretrain DeCUR in three multimodal scenarios: SAR-optical, RGB-DEM and RGB-depth. For SAR-optical, we use the SSL4EO-S12 dataset (Wang et al., 2022c) which consists of 250k multi-modal (SAR-multispectral) multi-temporal (4 seasons) image triplets with size 264x264. One random season is selected to generate each augmented view. For RGB-DEM, we conduct pretraining on the training set of GeoNRW dataset (Baier et al., 2020). The dataset includes orthorectified aerial photographs (RGB), LiDAR-derived digital elevation models (DEM) and open street map refined segmentation maps from the German state North Rhine-Westphalia. We

crop the raw 6942 training scenes to 111k patches with size 250x250. For RGB-depth, we use SUN-RGBD dataset which consists of 10335 RGBD pairs with various image sizes. Following Zhang et al. (2022), we preprocess the depth images to HHA format (Gupta et al., 2014).

**Data augmentations**   We follow common augmentations in the SSL literature (Grill et al., 2020) for optical and RGB images, and remove non-doable ones for specific modalities. Specifically, for SAR images, we use random resized crop (224 × 224), grayscale, Gaussian blur, and horizontal and vertical flip; for DEM images, we use random resized crop (224 × 224) and horizontal and vertical flip; for HHA images, we use random resized crop (224 x 224) and horizontal flip.

**Model architecture**   As a multimodal extension of Barlow Twins (Zbontar et al., 2021), each branch holds a separate backbone and a 3-layer MLP projector (each with output dimension 8192). DeCUR is trained on embedding representations after the projector, whose dimensions are separated to common and unique. We do a light grid search to get the best corresponding ratio. For SAR-optical, the percentage of common dimensions is 87.5%; for RGB-DEM and RGB-depth it is 75%. The backbones are transferred to downstream tasks. We use ResNet-50 (He et al., 2016) for all scenarios, with additional segformers (Xie et al., 2021) for RGB-Depth.

**Optimization**   We follow the optimization protocol of Barlow Twins (Zbontar et al., 2021) and BYOL (Grill et al., 2020), with default epochs 100 and a batch size of 256 (epochs 200 and batch size 128 for RGB-depth). The trade-off parameters $\lambda$ of the loss terms are set to 0.0051. Training is distributed across 4 NVIDIA A100 GPUs and takes about 30 hours on SSL4EO-S12, 4 hours on GeoNRW, and 6 hours on SUN-RGBD.

## 5   EXPERIMENTAL RESULTS

We evaluate DeCUR by pretraining and transferring to three common multimodal tasks: SAR-optical scene classification, RGB-DEM semantic segmentation, and RGB-depth semantic segmentation. We follow common evaluation protocols of self-supervised learning: linear classification (with frozen encoder) and fine-tuning. We report results for full- and limited-label settings, and both multimodal and missing-modality (i.e., only a single modality is available) scenarios.

### 5.1   SAR-OPTICAL SCENE CLASSIFICATION

We pretrain SAR-optical encoders on SSL4EO-S12 (Wang et al., 2022c) and transfer them to BigEarthNet-MM (Sumbul et al., 2021), a multimodal multi-label scene classification dataset with 19 classes. Simple late fusion is used for multimodal transfer learning, i.e., concatenating the encoded features from both modalities, followed by one classification layer. Mean average precision (mAP, global average) is used as the evaluation metric.

We report multimodal linear classification and fine-tuning results with 1% and 100% training labels in Table 1 (left). DeCUR outperforms existing cross-modal SimCLR-like contrastive learning by 2%-4.8% in most scenarios, while achieving comparable performance on fine-tuning with full labels. Notably, Barlow Twins itself works better than both SimCLR and VICReg (Bardes et al., 2021). Compared to BarlowTwins, we improve by 0.7% and 1.4% on linear evaluation and fine-tuning with 1% labels, and 2.2% and 0.2% with full labels.

Additionally, we report SAR-only results in Table 1 (right), as it is an essential scenario in practice when optical images are either unavailable or heavily covered by clouds. DeCUR outperforms other methods in most scenarios by a large margin, while achieving comparable performance on fine-tuning with full labels. In addition, DeCUR outperforms single-modal Barlow Twins pretraining by 2.1%-2.5% with 1% labels and 0.8%-2.1% with full labels, indicating that joint multimodal pretraining helps the model better understand individual modalities.

### 5.2   RGB-DEM SEMANTIC SEGMENTATION

We pretrain and evaluate RGB-DEM encoders on GeoNRW (Baier et al., 2020) for semantic segmentation (10 classes). We use simple fully convolutional networks (FCN) (Long et al., 2015) as

Table 1: SAR-optical transfer learning results on BigEarthNet-MM. Left: multimodal; right: SAR-only. We report linear classification and fine-tuning scores for training with both 100% and 1% labels. Rand. Init. represents random initialization, -cross represents cross-modal, -SAR represents SAR-only. Best per-column scores are marked in **bold**.

| SAR-optical | 1% labels | | 100% labels | |
|---|---|---|---|---|
| | Linear | Fine-tune | Linear | Fine-tune |
| Rand. Init. | 58.7 | 58.7 | 70.1 | 70.1 |
| Supervised | 77.0 | 77.0 | 88.9 | 88.9 |
| SimCLR-cross | 77.4 | 78.7 | 82.8 | 89.6 |
| CLIP | 77.4 | 78.7 | 82.8 | 89.6 |
| Barlow Twins | 78.7 | 80.3 | 83.2 | 89.5 |
| VICReg | 74.5 | 79.0 | 81.9 | 89.5 |
| DeCUR (ours) | **79.4** | **81.7** | **85.4** | **89.7** |

| SAR | 1% labels | | 100% labels | |
|---|---|---|---|---|
| | Linear | Fine-tune | Linear | Fine-tune |
| Rand. Init. | 50.0 | 50.0 | 54.2 | 54.2 |
| Supervised | 67.5 | 67.5 | 81.9 | 81.9 |
| SimCLR-cross | 68.1 | 70.4 | 71.7 | **83.7** |
| CLIP | 68.0 | 70.2 | 71.7 | 83.4 |
| Barlow Twins | 72.3 | 73.7 | 77.8 | 83.6 |
| VICReg | 69.3 | 71.9 | 74.1 | 83.6 |
| Barlow Twins-SAR | 71.2 | 73.3 | 77.5 | 81.6 |
| DeCUR (ours) | **73.7** | **75.4** | **78.3** | 83.7 |

the segmentation model, which concatenates the last three layer feature maps from both modalities, upsamples and sums them up for the prediction map. Similar to the classification task, linear classification is conducted by freezing the encoder, and fine-tuning is conducted by training all model parameters. Mean intersection over union (mIoU) is used as the evaluation metric.

Table 2: RGB-DEM transfer learning results on GeoNRW. Left: multimodal; right: RGB-only. We report linear classification and fine-tuning mIoU scores for training with both 100% and 1% labels.

| RGB-DEM | 1% labels | | 100% labels | |
|---|---|---|---|---|
| | Linear | Fine-tune | Linear | Fine-tune |
| Rand. Init. | 14.1 | 14.1 | 23.0 | 23.0 |
| Supervised | 22.1 | 22.1 | 44.0 | 44.0 |
| SimCLR-cross | 23.0 | 30.2 | 35.2 | 47.3 |
| CLIP | 22.8 | 28.8 | 35.0 | 46.7 |
| Barlow Twins | 31.2 | 33.6 | 43.0 | 48.4 |
| VICReg | 27.4 | 32.8 | 38.0 | 45.1 |
| DeCUR (ours) | **34.9** | **36.9** | **43.9** | **48.7** |

| RGB | 1% labels | | 100% labels | |
|---|---|---|---|---|
| | Linear | Fine-tune | Linear | Fine-tune |
| Rand. Init. | 14.2 | 14.2 | 18.5 | 18.5 |
| Supervised | 17.5 | 17.5 | 38.8 | 38.8 |
| SimCLR-cross | 20.1 | 25.9 | 29.6 | 42.5 |
| CLIP | 20.0 | 25.7 | 29.4 | 42.3 |
| Barlow Twins | 29.4 | 33.4 | 38.0 | 45.9 |
| VICReg | 23.7 | 28.7 | 32.4 | 41.6 |
| BarlowTwins-RGB | 28.6 | 32.6 | 36.2 | 45.7 |
| DeCUR (ours) | **31.4** | **34.5** | **43.9** | **46.5** |

We report multimodal linear classification and fine-tuning results with 1% and 100% training labels in Table 2 (left). Promisingly, DeCUR outperforms other methods in all scenarios by a large margin (up to 12.1% compared to CLIP). Meanwhile, we report RGB-only results in Table 2 (right), as in practice DEM data is not always available. Again DeCUR shows a significant improvement compared to others in all scenarios (up to 11.4% compared to CLIP).

## 5.3 RGB-DEPTH SEMANTIC SEGMENTATION

We pretrain RGB-depth encoders on SUN-RGBD (Song et al., 2015) and transfer them to SUN-RGBD and NYU-Depth v2 (Nathan Silberman & Fergus, 2012) datasets for semantic segmentation (37 and 40 classes, respectively). We transfer ResNet50 to simple FCN (Long et al., 2015) and Segformer (Xie et al., 2021) to the recent CMX (Zhang et al., 2022) model. We report single and multimodal fine-tuning results with mIoU and overall accuracy. As is shown in Table 3, when using simple segmentation models, DeCUR helps improve FCN over CLIP by 4.0% mIoU and 1.3% accuracy on SUN-RGBD, and 0.8% mIoU and 0.6% accuracy on NYU-Depth v2.

Promisingly, consistent improvements are observed by simply transferring the pretrained backbones to SOTA supervised mutimodal fusion models. Following the published codebase and without tuning any hyperparameter, we push CMX-B2 from mIoU 49.7% to 50.6% on SUN-RGBD dataset, and CMX-B5 from mIoU 56.9% to 57.3% on NYU-Depth v2 dataset.

## 6 ABLATION STUDIES

For all ablation studies, we pretrain ResNet-50 backbones on SSL4EO-S12 for SAR-optical and GeoNRW for RGB-DEM. Unless explicitly noted, we do fine-tuning on BigEarthNet-MM (SAR-optical) and GeoNRW (RGB-DEM) with 1% training labels.

Table 3: RGB-depth transfer learning results on SUN-RGBD (left) and NYU-Depth v2 (right).

| SUN-RGBD | modal | mIoU | Acc. |
|---|---|---|---|
| FCN (Long et al., 2015) | RGB | 27.4 | 68.2 |
| FCN (CLIP (Radford et al., 2021)) | RGB | 30.5 | 74.2 |
| FCN (DeCUR) | RGB | 34.5 | 75.5 |
| SA-Gate (Chen et al., 2020b) | RGBD | 49.4 | 82.5 |
| SGNet (Chen et al., 2021) | RGBD | 48.6 | 82.0 |
| ShapeConv (Cao et al., 2021) | RGBD | 48.6 | 82.2 |
| CMX-B2 (Zhang et al., 2022) | RGBD | 49.7 | 82.8 |
| CMX-B2 (DeCUR) | RGBD | **50.6** | **83.2** |

| NYUDv2 | modal | mIoU | Acc. |
|---|---|---|---|
| FCN (Long et al., 2015) | RGB | 29.2 | 60.0 |
| FCN (CLIP (Radford et al., 2021)) | RGB | 30.4 | 63.3 |
| FCN (DeCUR) | RGB | 31.2 | 63.9 |
| SA-Gate (Chen et al., 2020b) | RGBD | 52.4 | 77.9 |
| SGNet (Chen et al., 2021) | RGBD | 51.1 | 76.8 |
| ShapeConv (Cao et al., 2021) | RGBD | 51.3 | 76.4 |
| OMNIVORE (Girdhar et al., 2022) | RGBD | 54.0 | - |
| CMX-B5 (Zhang et al., 2022) | RGBD | 56.9 | 80.1 |
| CMX-B5 (DeCUR) | RGBD | **57.3** | **80.3** |

**Loss terms**   The ablation results about the components of our loss terms are shown in Table 4. We first remove both intra-modal training and modality decoupling, i.e., a cross-modal Barlow Twins remains. The downstream performance decreased as expected, as neither intra-modal information nor modality-unique information is learned. Then we remove intra-modal training and keep modality decoupling, which gives unstable performance change for different modality scenarios. This can be explained by the fact that without intra-modal training the unique dimensions can be randomly generated and are not necessarily meaningful. Finally, we remove modality decoupling and keep intra-modal training, which gives second best performance among the ablations. This confirms the benefits of intra-modal representations which can be a good complement to commonly learnt cross-modal representations. All of the above are below the combined DeCUR, proving the effectiveness of the total DeCUR loss.

Table 4: Ablation results on different loss components. *intra* corresponds to intra-modal training; *decoup.* corresponds to modality decoupling. We report mAP-micro score on BigEarthNet-MM for SAR-optical, and mIoU score on GeoNRW for RGB-DEM.

| | SAR-optical (mAP) | RGB-DEM (mIoU) |
|---|---|---|
| DeCUR (ours) | **81.7** | **36.9** |
| w/o intra&decoup. | 80.3 | 33.6 |
| w/o intra | 80.1 | 34.3 |
| w/o decoup. | 81.1 | 35.2 |

**Percentage of common dimensions**   We do a simple grid search based on downstream performance to find the best ratio between common and unique dimensions for SAR-optical and RGB-DEM respectively, as different modality combinations may have different representation overlaps. As is shown in Figure 3a, the best percentage of common dimensions is 87.5% for SAR-optical and 75% for RGB-DEM. This could be in line with the fact that there is more valid modality-unique information in orthophoto and elevation model than in optical and SAR (when the optical image is cloud-free). In both scenarios, the downstream performance increases and decreases smoothly along with the reduced percentage of common dimensions. Interestingly, there is no significant performance drop when decoupling up to 50% unique dimensions. This indicates the sparsity of the common embedding space.

**Number of projector dimensions**   Inherited from Barlow Twins (Zbontar et al., 2021), DeCUR also benefits from the increasing dimensionality of the projector. As can be seen from Figure 3b, DeCUR keeps improving with all output dimensionality tested.

**Effect of the projector**   Interestingly, DeCUR works well on the segmentation task even without the projector. As is shown in Figure 3b, removing the projector gives reasonable downstream performances, while adding it can further enhance the representations with a large number of dimensions.

## 7   DISCUSSION

In this section, we demonstrate an explainability analysis to interpret the multimodal representations learnt by DeCUR. We illustrate SAR-optical here, see Appendix for other multimodal scenarios.

**Cross-modal representation alignment**   To monitor the fact that each modality contains unique information that is difficult to integrate into a common space, we calculate the cross-modal align-

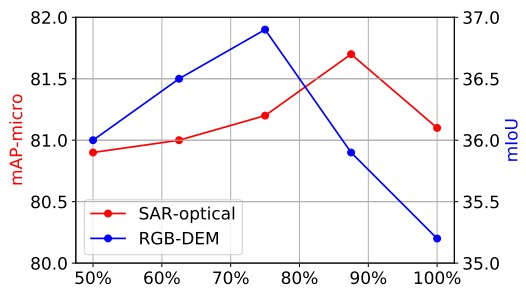 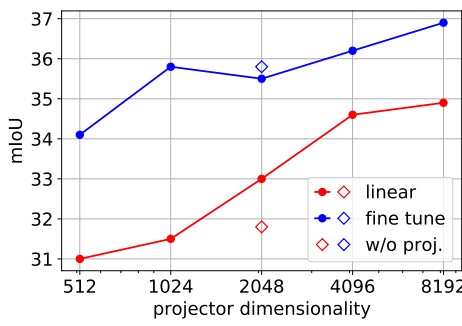

(a) Ablation results on the percentage of common dimensions. Best percentage is 87.5% for SAR-optical, and 75% for RGB-DEM.

(b) Effect of the existence and dimensionality of the projector. We report linear and fine-tuning results on GeoNRW dataset.

Figure 3: Ablation results on the percentage of common dimensions and the projector.

ment of every embedding dimension. This is done by counting the on-diagonal losses of the cross-correlation matrix $\mathcal{C}$:

$$\mathcal{L}_i = (1 - \mathcal{C}_{ii})^2 \tag{7}$$

where $i$ is the $i_{th}$ embedding dimension. The closer $\mathcal{L}_i$ to 0, the better the alignment of the two modalities in this dimension. We count the loss for all dimensions and plot the histogram of one random batch for both DeCUR and cross-modal Barlow Twins. The former explicitly decouples unique dimensions, while the latter assumes that all dimensions are common. As is shown in Figure 4a, the alignment loss remains high for a certain number of dimensions with cross-modal Barlow Twins. On contrary, by allowing the decorrelation of several dimensions (the loss of which moves to 1), the misalignment of common dimensions decreases. We further visualize such effects with t-SNE by clustering among the embedding dimensions. Contrarily to the common t-SNE setting that each input sample is one point, we make each embedding dimension one point. As Figure 1 shows, modality-unique dimensions are well separated, and common dimensions are perfectly overlapped.

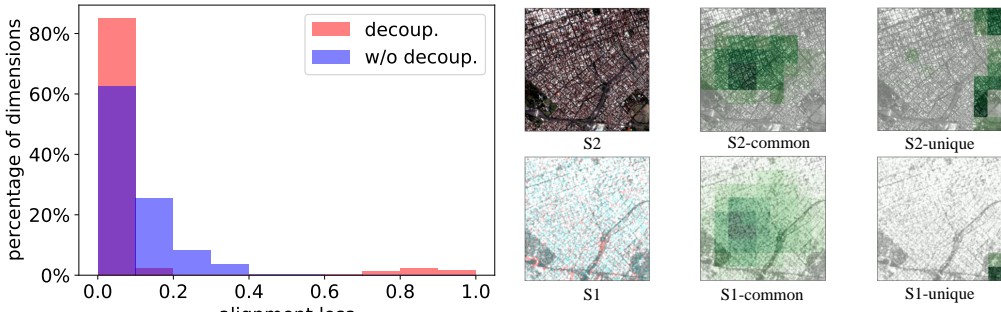

(a) SAR-optical representation alignment analysis.

(b) SAR-optical GradCAM visualization. S2 indicates optical, and S1 indicates SAR.

Figure 4: Cross-modal representation alignment (left) and spatial saliency visualization (right).

**Spatial saliency visualization** We use GradCAM (Selvaraju et al., 2017) to visualize the spatial saliency of input modalities corresponding to the common and unique embedding representations. For preparation, we average the common and unique dimensions as two single values output. Next, one-time backpropagation is performed w.r.t the corresponding output target (0 for common and 1 for unique) to get the GradCAM saliency map after the last convolutional layer. We then upsample the saliency maps to the size of the input. In total, one "common" and one "unique" saliency map are generated for each modality. We present one example for SAR-optical in Figure 4b, which shows an overlap in interest region for the common representations and tend to be orthogonal for the unique representations. See the appendix for more examples.

**Spatial saliency statistics** We further calculate the statistics of the common and unique highlighted areas for the whole pretraining dataset. We multiply the saliency maps between common and between unique for the two modalities, take the logarithm, and normalize the results of each patch to 0 to 1. In other words, for each pair of images, we calculate one score for common area similarity and one for unique area similarity. We thus get one histogram for common and one for unique as shown in Figure 5a. Though not significant, the histograms show a slight trend of unique scores being more towards 0 than common scores, indicating that the interesting areas of modality-unique representations tend to be more orthogonal than common representations which tend to overlap.

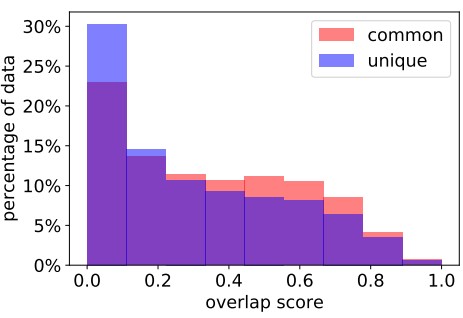

(a) SAR-optical spatial saliency statistics.

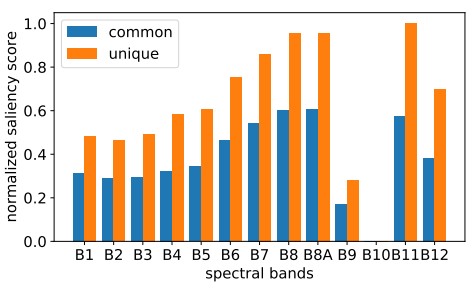

(b) Spectral saliency statistics for the 13 optical bands. A higher saliency score indicates higher importance.

Figure 5: Spatial saliency statistics (left) and spectral saliency statistics (right).

**Spectral saliency statistics** The insignificant difference in spatial saliency statistics are as expected, because the image-level semantics can not only be presented at spatial domain, but also other aspects such as the spectral domain for multispectral images. Therefore, we use Integrated Gradients (Sundararajan et al., 2017) to perform saliency analysis back to the input and count statistics over spectral bands in optical images. We don't use GradCAM here as it tends to lose class discriminability in shallow layers (Selvaraju et al., 2017). An importance score is assigned to each input feature by approximating the integral of gradients of the output (the preparation is the same as spatial saliency above) w.r.t. the inputs. We then average the importance scores of each band to get spectral saliency for both common and unique representations. We normalize the scores and do statistics over the whole SSL4EO-S12 dataset, and plot the histograms in Figure 5b. The figure confirms the bigger influence of the spectral information on optical-unique representations. Meanwhile, the band-wise importance distribution is promisingly consistent with the domain knowledge: 1) near-infrared bands (B5-B8A, including vegetation red edge) are very important; 2) red (B4) is more important than blue (B2); 3) water vapour (B9) and cirrus (B10) are strongly related to atmosphere and thus less important for land surface monitoring; etc.

## 8 CONCLUSION

We presented DeCUR, a simple yet insightful multimodal self-supervised learning method. We introduced the idea of modality decoupling and intra-modal representation enhancing which can be implemented as a simple extension of Barlow Twins. Extensive experiments on three common multimodal scenarios prove the effectiveness of DeCUR. Moreover, we conduct a systematic explainability analysis to interpret the proposed method. Our results suggest that modality-decoupling bears great potential for multimodal representation learning.

Future work considers more complex multimodal scenarios, where one modality may contain more unique information than the other, and different samples may not have the same information distribution. By allocating the same number of unique dimensions for both modalities across the dataset, DeCUR simplifies the decoupling design but does not address the sparsity of the less informative modality's latent space and the imbalance across different instances. Another limitation is that a grid search is needed for the best percentage of common dimensions, which can be costly on a huge dataset. While a general percentage of around 80% can achieve reasonable performance in our tested scenarios, a more efficient discovering strategy is to be explored. Nevertheless, we believe this work serves as a valuable starting point for future research, such as the exploration of adaptive decoupling strategies and integrating more modalities in a unified framework.

## REPRODUCIBILITY STATEMENT

To ensure the reproducibility of this paper, we provide: 1) pseudo codes for the DeCUR algorithm 1 and the explainability analysis 2; 2) implementation details for both pretraining and transfer learning on all datasets in section 4 and the appendix B; 3) source codes in the supplementary material.

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

# A  ALGORITHM

---

**Algorithm 1:** PyTorch-style pseudocode for DeCUR.

```
# f1,f2:  encoder networks
# BN: batch normalization
# N,K: batch size and embedding dimension
# on_diag, off_diag:  on- and off-diagonal elements of a matrix

# loss function for common and intra-modal
def loss_c(C, lambda):
   l_on = (on_diag(C)-1).pow(2).sum()
   l_off = off_diag(C).pow(2).sum()
   return l_on + lambda x l_off

# loss function for unique
def loss_u(C, lambda):
   l_on = on_diag(C).pow(2).sum()
   l_off = off_diag(C).pow(2).sum()
   return l_on + lambda x l_off

# training
for x1,x2 in loader:  # load a batch pairs
   # two augmented views for each modality
   x1_1, x1_2 = augment1(x1)
   x2_1, x2_2 = augment2(x2)
   # compute embeddings and normalize
   z1_1, z1_2 = BN(f1(x1_1)), BN(f1(x1_2))
   z2_1, z2_2 = BN(f2(x2_1)), BN(f2(x2_2))
   # cross-correlation matrices
   C1 = z1_1.T @ z1_2 / N # KxK
   C2 = z2_1.T @ z2_2 / N # KxK
   Cm = z1_1.T @ z2_1 / N # KxK
   Cc = Cm[:K_c,:K_c] # KcxKc
   Cu = Cm[K_c:,K_c:]  # KuxKu
   # calculate losses
   L1 = loss_c(C1,lmb1) # intra-modal M1
   L2 = loss_c(C2,lmb2) # intra-modal M2
   Lc = loss_c(Cc,lmbc) # cross-m.  common
   Lu = loss_u(Cu,lmbu) # cross-m.  unique
   loss = L1 + L2 + Lc + Lu # total loss
   # optimization
   loss.backward()
   optimizer.step()
```

---

# B  ADDITIONAL IMPLEMENTATION DETAILS

**SAR-optical pretraining**  SSL4EO-S12 (Wang et al., 2022c) dataset is used for SAR-optical pre-training: Sentinel-1 GRD (2 bands VV and VH) and Sentinel-2 L1C (13 multispectral bands). The pixel resolution is united to 10 meters. Following the original settings, we compress and normalize the optical data to 8-bit by dividing 10000 and multiply 255; for SAR data, we cut out 2% outliers for each image and normalize it by band-wise mean and standard deviation of the whole dataset.

Standard ResNet50 is used as the encoder backbone, of which the first layer is modified to fit the input channel number. The projector is a 3-layer MLP, of which the first two layers include Linear, BactchNorm and ReLU, and the last one includes only a linear layer.

We use the LARS (You et al., 2017) optimizer with weight decay 1e-6 and momentum 0.9. We use a learning rate of 0.2 for the weights and 0.0048 for the biases and batch normalization parameters. We reduce the learning rate using a cosine decay schedule (Loshchilov & Hutter, 2016) (no warm-up periods). The biases and batch normalization parameters are excluded from LARS adaptation and weight decay.

**RGB-DEM pretraining**   The training split of GeoNRW (Baier et al., 2020) dataset is used for RGB-DEM pretraining: aerial orthophoto (3 bands RGB) and lidar-derived digital elevation model (1 band heights). The pixel resolution is 1 meter. We use standard ResNet50 without modifying the input layer (i.e., we duplicate DEM image to 3 channel). Other model architecture and optimization protocols are the same as SAR-optical pretraining.

**RGB-depth pretraining**   SUN-RGBD (Song et al., 2015) dataset is used for RGB-depth pretraining: indoor RGB and depth images. Following Zhang et al. (2022), we preprocess the depth images to HHA format (Gupta et al., 2014). We use standard ResNet50 and segformer-B2/B5 as the backbones. For segformer backbones, we use AdamW optimizer and a learning rate of 1e-4.

**SAR-optical transfer learning**   We evaluate SAR-optical pretraining on BigEarthNet-MM (Sumbul et al., 2021) dataset for the multi-label scene classification task. We compress and normalize the optical images to 8-bit by dividing 10000 and multiply 255; for SAR images, we cut out 2% outliers for each image and normalize it by band-wise mean and standard deviation of the whole dataset. As the optical data of BigEarthNet-MM is Sentinel-2 L2A product (12 bands), we insert one empty band to match the pretrained weights (13 bands). We use common data augmentations including RandomResizedCrop (scale 0.8 to 1) and RandomHorizontalFlip.

Standard ResNet50 is used as the encoder backbone for each modality, of which the first layer is modified to fit the input channel number, and the last layer is modified as an identity layer. The encoded features are concatenated, followed by a fully connected layer outputting the class logits. The encoders are initialized from the pretrained weights. For linear classification, the encoder weights are frozen and only the last classification layer is trainable; for fine-tuning, all weights are trained.

We optimize MultiLabelSoftMarginLoss with batchsize 256 for 100 epochs. We use the SGD optimizer with weight decay 0 and momentum 0.9. The learning rate is 0.5 for linear classification, and 0.05 for fine-tuning. We reduce the learning rate by factor 10 at 60 and 80 epochs.

**RGB-DEM transfer learning**   We evaluate RGB-DEM pretraining on GeoNRW dataset for the semantic segmentation task. We use common data augmentations including RandomResizedCrop (scale 0.2 to 1) and RandomHorizontalFlip.

Fully convolutional networks (FCN) (Long et al., 2015) with standard ResNet50 backbone for each modality is used as the segmentation model. The last three feature maps from both modalities are concatenated and upsampled to the input size. They are further followed by 1x1 convolution outputting three segmentation maps, which are added together to form the final output map. The encoders are initialized from the pretrained weights. For linear classification, the encoder weights are frozen; for fine-tuning, all weights are trainable.

We optimize CrossEntropyLoss with batchsize 256 for 30 epochs. We use the AdamW optimizer with weight decay 0.01. The learning rate is 0.0001 for both linear classification and fine-tuning.

**RGB-depth transfer learning**   We evaluate RGB-depth pretraining on SUN-RGBD and NYU-Depth v2 datasets for the semantic segmentation task. We use common data augmentations including RandomResizedCrop and RandomHorizontalFlip.

FCN with ResNet50 backbones are used as the segmentation model for single-modal RGB semantic segmentation. We optimize CrossEntropyLoss with batchsize 8 for 40k iterations. We use the SGD optimizer with weight decay 1e-5. The learning rate is 0.01 with polynomial decay for fine tuning.

CMX (Zhang et al., 2022) with segformer (Xie et al., 2021) backbones are used as the segmentation model for RGBD semantic segmentation. We follow the same settings of CMX for SUN-RGBD and NYU-depth v2 datasets.

## C  ADDITIONAL EXPERIMENTAL RESULTS

**Robustness of common dim. percentage**  We do a grid search to find the best percentage of common dimensions. However, this is built upon the fact that the total embedding dimension is 8192. Will the best percentage change when the embedding dimensionality changes? To answer this question, we repeat the search with a total of 512 embedding dimensions on SAR-optical and RGB-DEM datasets. As is shown in Figure 6, the best percentage of common dimensions are interestingly the same for both the small embedding space and the big embedding space.

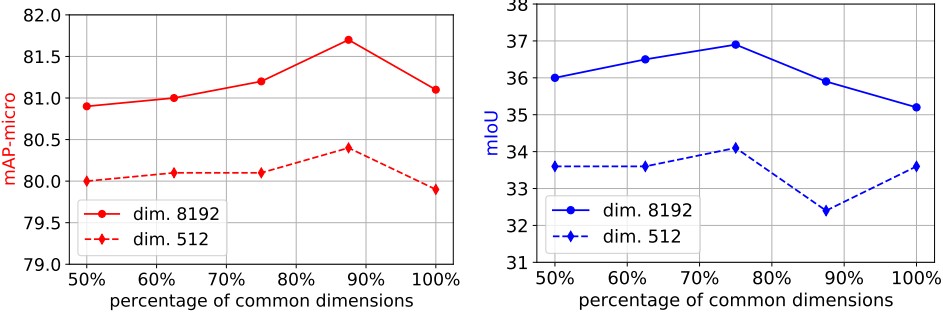

Figure 6: Ablation results on the percentage of common dimensions with different total dimensionalities. Best percentages are consistent with different scenarios (left: SAR-optical multilabel classification, right: RGB-DEM semantic segmentation).

**BigEarthNet-MM ablation on proj. dim.**  Figure 3b in the main paper shows the effect of the projector on semantic segmentation GeoNRW dataset. As a supplement, we report the results on scene classification BigEarthNet-MM dataset in Figure 7. While the ablation on projector dimensions is consistent with GeoNRW, removing the projector hurts the performance significantly.

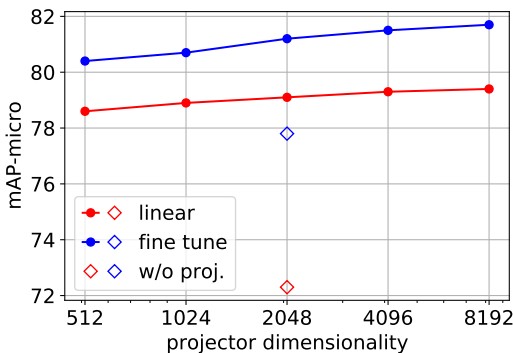

Figure 7: Effect of the existence and dimensionality of the projector. We supplement with the results on BigEarthNet-MM dataset.

# D  EXPLAINABILITY ANALYSIS

## D.1  IMPLEMENTATION PSEUDOCODE

For better understanding of our explainability implementation, we provide a united pseudocode in Algorithm 2.

---

**Algorithm 2:**  Pseudocode for DeCUR explainability.

```
# f1,f2:  encoder networks
# BN: batch normalization
# N,K: batch size and embedding dimension
# on_diag:  on-diagonal elements of a matrix
# IG: Integrated Gradients

# 1.Cross-modal representation alignment
def alignment_histogram(z1, z2):
   C = z1.T @ z2 / N # KxK
   losses = (on_diag(C)-1).pow(2) # Kx1
   return histogram(losses, range=(0,1))

# 2.Representation visualization
def tsne_vis(z1, z2):
   feature = torch.cat((z1,z2),-1) # Nx2K
   feature = feature.permute(1,0) # 2KxN
   return tsne(feature, n_components=2)

# 3.Spatial saliency visualization
def gradcam_vis(x1):
   z1 = BN(f1(x1)) # NxK
   z1_c = z1[:,:Kc].mean(dim=-1) # Nx1
   z1_u = z1[:,Kc:].mean(dim=-1) # Nx1
   out1 = torch.cat((z1_c,z1_u),-1) # Nx2
   gc1 = LayerGradCam(f1, f1.last_conv2)
   attr1_c = gc1.attribute(x1,target=0) # Nx7x7
   attr1_u = gc1.attribute(x1,target=1) # Nx7x7
   return upsamp(attr1_c), upsamp(attr1_u)
   # Nx224x224, Nx224x224

# 4.Spatial saliency statistics
def gradcam_stat(x1,x2):
   att1_c, att1_u = gradcam_vis(x1)
   att2_c, att2_u = gradcam_vis(x2)
   mul_c = norm(att1_c) x norm(att2_c) # Nx1
   mul_u = norm(att1_u) x norm(att2_u) # Nx1
   return mul_c, mul_u

# 5.Spectral saliency statistics
def IG_stat(x1):
   # define "IG_vis" similar to gradcam
   att1_c, att1_u = IG_vis(x1) # NxCx224x224
   imp_c = att1_c.mean(dim=(0,2,3)) # NxC
   imp_u = att1_u.mean(dim=(0,2,3)) # NxC
   return imp_c, imp_u
```

---

## D.2 ADDITIONAL EXAMPLES

Below we show some additional explainability examples. Note that the decoupling and matching results depend on the samples. Specifically, some images have strong overlap between modalities (potentially more common dimensions) while the others tend to be more orthogonol (potentially more unique dimensions, decoupling helps more).

**Cross-modal alignment histograms**    See Figure 8.

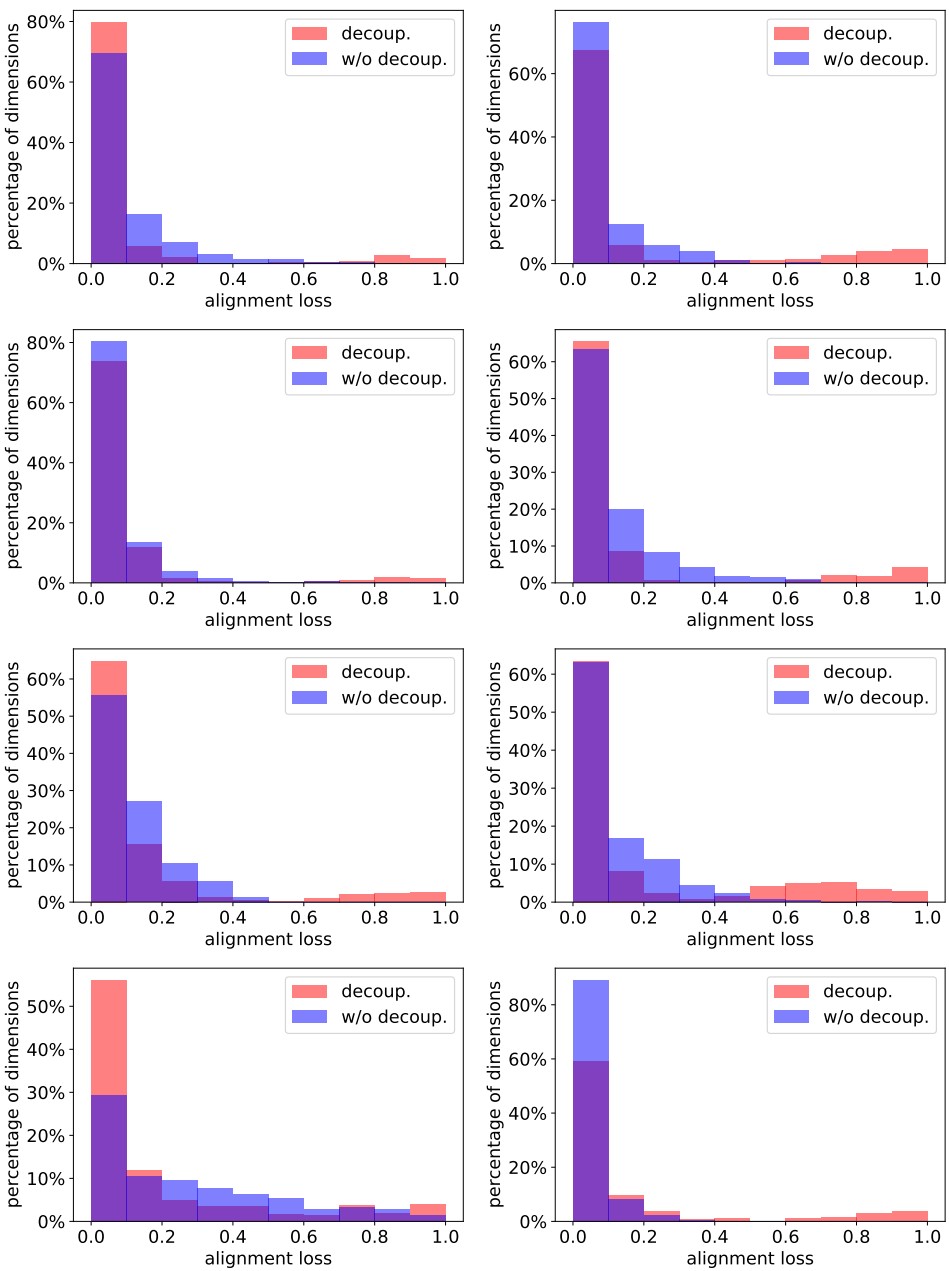

Figure 8: Cross-modal representation alignment histograms of 4 batches of samples. Left: SAR-optical; right: RGB-DEM.

**t-SNE representation visualization** See Figure 9.

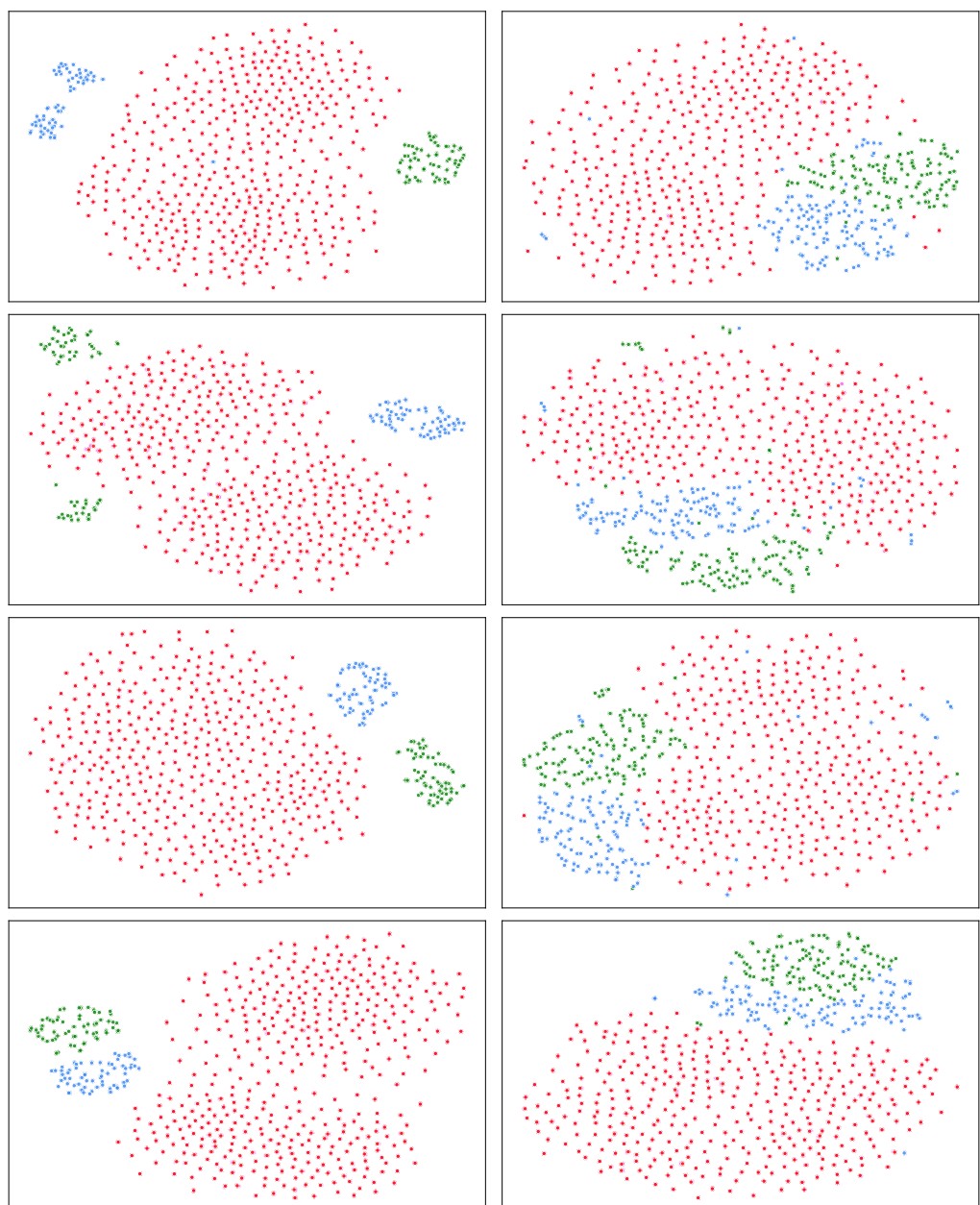

Figure 9: t-SNE representation visualization of 4 batches of samples. Left: SAR-optical; right: RGB-DEM.

**Spatial saliency visualization**  See Figure 10.

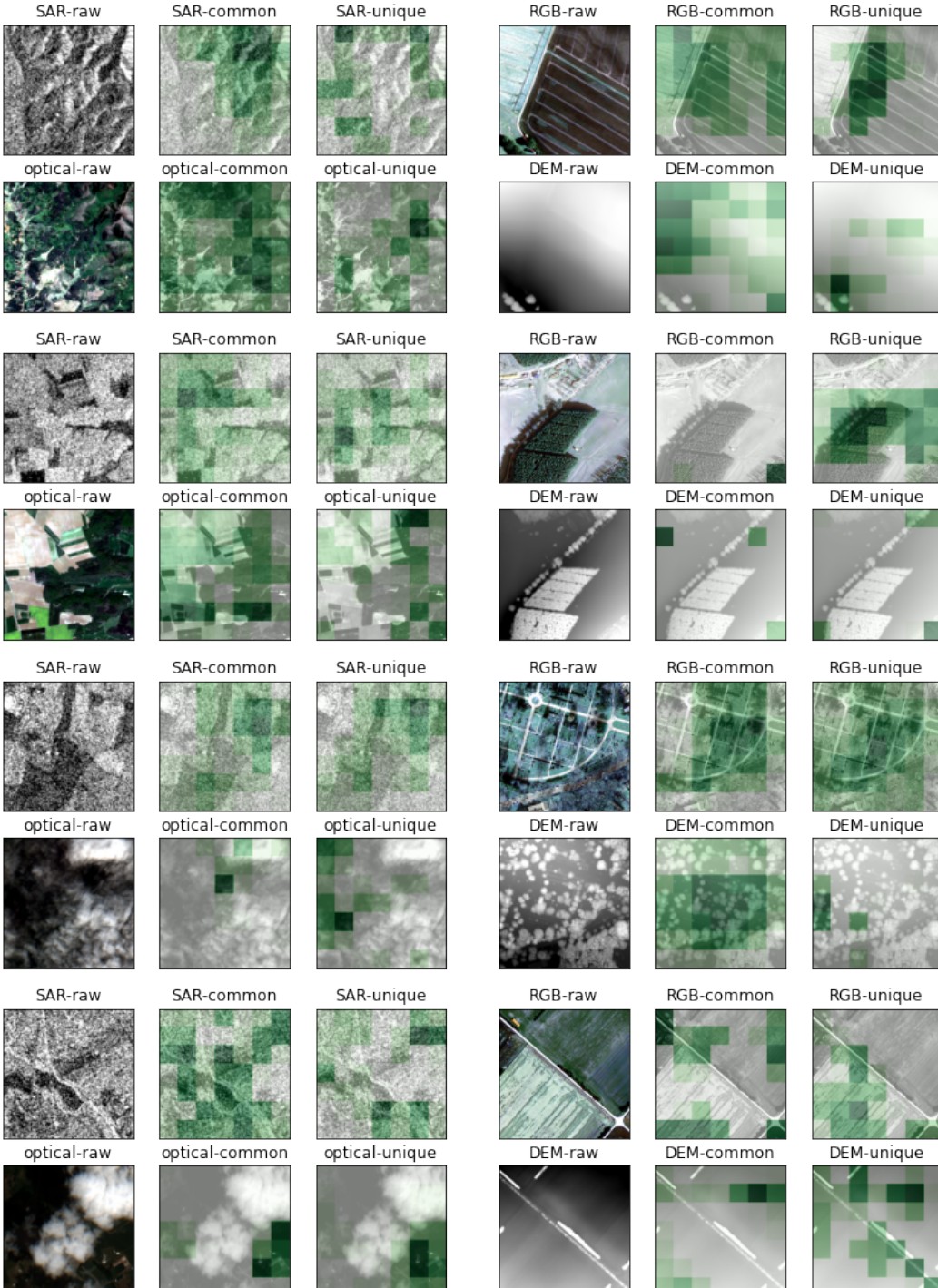

Figure 10: GradCAM visualization of 4 sample pairs. Left: SAR-optical; right: RGB-DEM.

