# OpenReview forum: "DeCUR: decoupling common & unique representations for multimodal self-supervision"
_ICLR.cc/2024/Conference — Submitted to ICLR 2024_

### Official Review · Reviewer_FCpx · 2023-10-20

**Soundness:** 2 fair
**Presentation:** 2 fair
**Contribution:** 2 fair
**Rating:** 5
**Confidence:** 4

**Summary:**

This paper proposes a self-supervised learning method for radar-optical, RGB-elevation, and RGB-depth joint data understanding. The design is simple, by disentangling modality factors of and between modalities. Benefiting from the proposed pretraining framework, the performance improvements on the downstream tasks are impressive.

**Strengths:**

* This paper is focused on practical multi-sensor complementary. It's a meaningful research topic.

* Performance improvements on 3 multimodal scenarios are impressive.

* The authors also provide a detailed analysis of their proposed method. It's useful to the community and inspiring for follow-up works.

**Weaknesses:**

* This paper lacks significant references, specifically, Omnivore [CVPR' 22] and ImageBind [CVPR'23] play an important role in this multimodal learning problem. However, neither was discussed in this paper.

* Technical contribution. It seems that the proposed DeCUR is the combination of CLIP [ICML'21] and SimCLR [ICML'20]. Please highlight the special design and insights.

**Questions:**

See Weakness.

---

> ### Author Response · Authors · 2023-11-14
>
> Thanks for your important feedback to help us improve the manuscript! We appreciate your positive view of our topic, the experiments and the analysis. We'll provide detailed responses below.
>
> **1. References.**
>
> Thanks for the suggestion! These two references are indeed important works in multimodal learning. We were a bit more focused on the story of dual-modal modality alignment and modality decoupling when introducing related works. We will add them to the revised manuscript. Specifically we will also add a comparison of NYUD-v2 segmentation result which was reported by OMNIVORE.
>
> **2. Technical contribution compared to CLIP and SimCLR.**
>
> Thanks for the question and we will explain the details in the following. SimCLR is a famous contrastive learning method that was originally designed for images only; CLIP shares with SimCLR a similar loss design (the contrastive InfoNCE loss) while extending to vision-language pretraining.
>
> Firstly, our method is conceptually different from CLIP and SimCLR, but rather inspired by BarlowTwins [ICML21] and VICReg [ICLR22]. Instead of contrasting negative samples along the batch dimension, we conduct redundancy reduction along the feature dimension. Assume we have the features of two input views $Z_1$, $Z_2$ with shape (B,D), where B is the batch size and D is the feature dimension.
> - CLIP/SimCLR calculates the cross-view cosine similarity, resulting in a similarity matrix with shape (B,B):
>
> $C = Z_1 \times {Z_2}^T$
>
> This matrix is optimized to be close to the Identity, thus each sample is only similar to its other view but dissimilar to other samples in this batch.
> - On the other hand, DeCUR/BarlowTwins calculates the cross-view cross-correlation matrix with shape (D,D):
>
> $C = {Z_1}^T \times Z_2$
>
> This matrix is optimized to be close to the Identity, thus each pair of views are similar to each other, while the redundancy between the feature dimensions is minimized. Similar to BarlowTwins, we do not rely on contrasting negative samples.
>
>
> Secondly about our contribution, CLIP/SimCLR/BarlowTwins (along with many other multimodal representation learning algorithms) tend to optimize the two views to be perfectly aligned while not considering modality-unique representations. This is too strict in practice, because one modality may have unique information that the other modality doesn’t have. Forcing the model to learn only the common information can lead to information loss. Therefore, our technical contribution lies in a novel design to decouple the common and unique representations during pretraining. We propose to separate the total embedding dimensions into common/unique ones, and make the common/unique dimensions aligned/de-aligned between two modalities. To avoid the collapse of the unique dimension, we make the unique dimensions aligned between two views within each single modality. The full design can be implemented based on the cross-correlation matrix, thus we believe it’s simple and efficient.
>
> **Summary**
>
> We hope the above responses address your concerns. We will also highlight our contributions in the revised manuscript for better understanding. Looking forward to your further feedback!

---

### Official Review · Reviewer_7Nuf · 2023-10-29

**Soundness:** 1 poor
**Presentation:** 2 fair
**Contribution:** 2 fair
**Rating:** 3
**Confidence:** 4

**Summary:**

Based on Barlow Twins, this paper proposes DeCUR, aiming to decouple common and unique representations in multimodal self-supervised learning. Specifically, the DeCUR splits the output embedding into common and unique parts. The common parts are used for common Barlow Twins losses, while the similarities of unique parts between modalities are trained to be zero. The authors evaluate the DeCUR on multiple scenarios and provide further analysis on the effect of the DeCUR.

**Strengths:**

1: The DeCUR is easy to understand and reproduce.

2: Experimental results show that the DeCUR can outperform some previous methods.

3: The authors conduct extensive explainability analysis for the interpretability of DeCUR.

**Weaknesses:**

### Method

1: **Forcing the correlation score of unique embeddings to be zero is not convincing.** As shown in Figure 2 and the methodology, the DeCUR splits the original output embedding into common and unique parts for each modality. The similarities of paried unique parts between modalities are **required to be zero**, while they **belong to the same instance** across different modalities. In other words, it means partial features of an instance from two modalities need to be mutually exclusive. Why can such training objectives make the model learn meaningful features? Could the authors provide further explanations? In the overall training loss, the common parts are trained to be aligned. So, will only the shared common features contribute to the representation ability, and will the unique parts be trained to be a trivial solution and meaningless?

### Experiments

1: **Missing comparisons for the main results.** In the introduction, the authors show the benefits of Barlow twins, like the fact that they don't need a large batchsize. A similar work, Mocov2 [1] (released in 2020), performs contrastive learning for self-supervised learning while reducing the necessity of a large batch size, which can also be included for comparison. In Table 2, the authors didn't compare the CLIP with the RGB-only scenario.

2: **Missing ablation studies of the effect of the common and unique features.** The decoupling is one of the main contributions in this work, while the analysis of the independent effects of the common and unique features is missed in the experiments part. What will the performance be if we only use the common or unique features for downstream tasks?

### Analysis

1: **The T-SNE visualizations.** In Figure 1, the common features of all the modalities are the same color and shape, which makes it hard for me to judge whether the common parts are well aligned (the authors only show one case using an enlarged circle).

 2: **The Cross-modal representation alignment analysis isn't convincing.** The authors calculate the correlation score of every corresponding dimension across two modalities' embeddings to illustrate the effect of feature decoupling. In other words, the authors judge the methods based on the assumption that the i-th dimension of modality A's embedding is paired with the i-th dimension of modality B's embedding. However, there is no guarantee that such an assumption is established since the alignment performs at the image-level rather than the channel-level.


Overall, the decoupling module isn't technically sound to me, and the experiments with analysis also aren't convincing. Therefore, I vote for rejection at this time.

[1] Chen, Xinlei, et al. "Improved baselines with momentum contrastive learning." arXiv preprint arXiv:2003.04297 (2020).

**Questions:**

Please refer to the weakness part.

---

> ### Author Response · Authors · 2023-11-14
> **Author response to the reviewer (P1)**
>
> Thanks for the constructive comments and important feedback! We really appreciate your detailed concerns and we'll explain/update point to point in the following.
>
> ### **Method**
>
> **1. The meaningfulness of decorrelating the unique dimensions.**
>
> Thanks for raising this question, which is one key point of this work and we'll discuss in detail. First of all, it is a correct understanding that one objective of our method is “partial features of an instance from two modalities need to be mutually exclusive”. This is because, very often one modality contains its modality-unique information that can not be extracted from the other modality, and vice versa. Such information, in one extreme case, can be simply the modality name (just an example for illustration). Therefore, even if the two modalities describe the same scene in general (the same “instance” as noted in your question), there are detailed aspects where they are not common. The total information of a pair of modalities consist of the common information between both, and the unique information from each modality respectively. If we do not decouple such unique information, we will force the model to learn only the common information, and some unique information can get lost during training.
>
> Then comes a following question: why is the unique information helpful? Take a pair of RGB/thermal-infrared images of one animal in the forest for example. The common information of the two modalities is only the animal, while the unique information from RGB can provide information about the environment's color/texture and the animal’s appearances, and the unique information from thermal-infrared can provide information about the environment and in-body temperature distribution. With the help of the unique information, we can have a fine-grained understanding of the scene beyond one rough idea. A very recent work FACTORCL [NeurIPS23] shares a very similar concept with us, proposing to learn self-supervised multimodal representations to capture both shared and unique information.
>
> Finally the question about the meaningfulness of decorrelating the unique dimensions. This is indeed one key aspect in our method. If we simply decorrelate the unique dimensions cross-modal, these dimensions can definitely collapse to a trivial solution “randomly decorrelated”. To avoid such collapse, we introduce intra-modal training where the unique dimensions contribute to the similarity between two views within one modality. In this way, the unique dimensions need to be meaningful. In fact, we believe this is a nice design which brings another advantage of learning also intro-modal representations to further boost the general performance.
>
> ### **Experiments**
>
> **1. Missing comparison of MoCov2 and CLIP for the main results.**
>
> MoCov2 by design requires momentum update of the encoder, which means the two encoders for different views are essentially the same. This makes it not applicable in our multimodal setting, where we train separate encoders for different modalities. This is also the case for many other self-supervised learning algorithms like SimSiam, BYOL, DINO which either have the same encoder or momentum encoder. We will mention this in the updates.
>
> We didn’t report CLIP in SAR-only (table1) and RGB-only (table2) because they perform similarly to SimCLR (also implementation-wise similar) as shown in SAR-RGB and RGB-DEM. Another minor reason is that we’d like to make the two tables’ space nicely aligned:) But yes we will add them in the revised paper for completeness.
>
> **2. Ablation of the independent common and unique features.**
>
> This is actually one interesing point. Both common and unique features are important for downstream tasks since they are decoupled and the combination of them forms the total information. Therefore, we expect the performance will naturally decrease if we only use common or unique parts for downstream tasks. But we agree it will be more clear and sound if we have such ablation. We have conducted a small ablation experiment as shown below:
>
> | Linear probing on BigEarthNet-MM with 1% labels              |  mAP (%) |
> |:-------------:|:--------------------------------------------------------:|
> | common&unique |                           79.4                           |
> |  common-only  |                           72.6                           |
> |  unique-only  |                           56.8                           |
>
> where "common&unique" is our reported result in the paper. It clearly shows that both common and unique parts are important for the downstream task.

---

> > ### Author Response · Authors · 2023-11-14
> > **Author response to the reviewer (P2)**
> >
> > ### **Analysis**
> > **1. t-SNE visualization in Figure 1.**
> >
> > Thanks for pointing out! As shown in the zoomed view of one dimension example, the common parts are “super well” aligned that they almost fully overlap. The colors from two modalities are in fact different but the visualization here isn’t so distinguishable. Sorry for the confusion here. We will update the figure with different markers and colors for better visualization.
> >
> > **2. Cross-modal representation alignment.**
> >
> > The alignment is indeed performed in dimension-level, i.e., we align each dimension of feature vector from one modality to the corresponding dimension of feature vector from the other modality. This is based on our loss design. Assuming the feature vectors have shape (B,D) with B is the batch size and D is the number of feature dimension, the correlation matrix would have shape (D,D). We then push the correlation matrix close to the Identity, i.e., the correlation between corresponding dimensions should be 1. With this objective design, we force the feature dimensions align with each other between the two modalities. This isn't contradictory with image-level alignment, since the total loss is the combination of all dimensions in the latent space.
> >
> > ### Summary
> > In summary, we hope the above responses answer your questions and clarify potential misleadings. We will also update the paper to make things clearer. Thanks again for your detailed comments, we look forward to your further feedback!

---

### Official Review · Reviewer_KmGU · 2023-10-29

**Soundness:** 3 good
**Presentation:** 3 good
**Contribution:** 2 fair
**Rating:** 5
**Confidence:** 3

**Summary:**

This paper proposes a multimodal representation learning approach, that aims at decoupling common and unique features from each modality.

**Strengths:**

+ The paper is well organized and clearly presented.
+ The proposed approach is simple, concise and efficient. It makes sense to divide the feature channels as common and unique, then enforce redundancy reduction through the resulting cross-correlation matrices.
+ Ablation study in Section 6 and analysis in Section 7 provide good insight.

**Weaknesses:**

+ The novelty seems limited as the major approach is an extension of Barlow Twins to the multimodal setting.

+ Figure 3 (a): It is good that the authors present the ablation study on the percentage of common dimensions. In a lot of multimodal scenarios, modality can be unbalanced -- some modality contain more common features while others contain less. I wonder if it makes more sense to set different percentage to different modalities. Another problem related to this is that the proposed approach seems to require grid search of this ratio to get the best performance. I assume that by changing the percentage of common dimension, retraining of the whole model is needed? This results in computational burdens, and stand as a disadvantage.

+ I wonder if the discovered modality-common and unique features can be utilized in some multimodal robustness settings, besides visualization. Say if one of the modality gets perturbed by Gaussian noise / there is a certain probability that one of the modality is missing during inference time, I expect the learned common features would be beneficial in this scenario while modality unique features from the perturbed modality would diminish the performance.

+ The datasets used for experiments, while being multimodal, are essentially all image modalities. Could the authors comment briefly about the applicability of the proposed approach to other modalities (say audio, text)?

**Questions:**

See weaknesses. I'm happy to increase my score if the authors could address my questions in the rebuttal.

---

> ### Author Response · Authors · 2023-11-14
> **Author response to the reviewer (P1)**
>
> Thanks for the constructive comments and suggestions! We are happy that you like our idea and are willing to increase the score! We provide point-to-point responses in the following.
>
> **1. Novelty as a multimodal extension of BarlowTwins.**
>
> We agree and actually want to highlight that our method is extremely simple:) We believe this simplicity is in fact one advantage of our method given the “complicated” optimization goals:
> - cross-modal representation alignment (this itself would be a simplest extension of BarlowTwins);
> - decoupling modality-specific unique representations;
> - avoiding the collapse of unique representations being trivial;
> - intra-modal cross-view alignment to enhance single-modal representations.
>
> All of the above goals can be connected together with the simple cross-correlation matrix. Notably, as mentioned in Figure 2 and Section 3, the 3rd goal (avoiding dimension collapse) can be naturally achieved together with the 4th goal (intra-modal learning). Such a highly-united form of different objectives makes the training easier and more efficient. As also mentioned in the related work, we did also consider VICReg [ICLR22] (an improved version of BarlowTwins), which would however break such a uniform design and introduce more hyparameters.
>
> **2. Unbalanced multimodal scenarios and the search for percentage ratio.**
>
> This is a very good point and is exactly what we want to further explore in the next research. I'd like to discuss a bit more in detail here. In multimodal scenarios, it's indeed common to encounter unbalanced modalities where one modality may contain more features that are common across modalities, while another may have a larger proportion of unique features.
>
> Let’s first simplify it in our dual-modal setting, where the amount of common information between two modalities should be the same. (*This is because of the definition of “common”, if the modalities>2 then A can have more common in B than in C. We leave such more complicated scenarios for future work.*) However, the amount of unique information can be different. In other words, the total amount of information can be different compared to each other. Then, there are two aspects regarding the balance of common/unique features:
> - in general (across the whole dataset), one modality has more unique information than the other;
> - for one specific pair, one modality has more unique information than the other.
>
> In this paper we mainly address the first aspect. As you mentioned, we empirically test the best percentage of common dimensions, and allocate the same amount of remaining dimensions for unique information for both modalities. Therefore, the total dimensions are “sufficient and unnecessary”, meaning for one less informative modality the unique space should be sparse. Our current design does not specifically deal with that, leading to the first limitation.
>
> For the second aspect where for specific instances the balance can be different, our current design also doesn’t specifically deal with it. This is to some extent similar as in common contrastive learning where "other samples in a batch or a queue" are not necessarily negatives. We assume the information balance holds statistically within a large dataset, but this is indeed the second limitation of this work.
>
> Finally, an empirical grid search for the best percentage is indeed a third limitation of our method. It is true that we have to retrain the whole model for a best ratio. We have to admit this is not very elegant, even though a rough estimation of the search space can be guided by human knowledge. Nevertheless, as shown in Figure 3 (a), starting from a bit less than 100% can always improve. A ratio of around 80% could be a sweet point in general. A more efficient way to get this ratio is to be explored.
>
> In summary, we think these limitations are also good starting points for follow-ups. In our future work, we consider exploring adaptive methods to determine the proportion of common and unique features across multiple modalities. Thanks again for raising this valuable point and we think it's also nice to update such discussion of limitations in the paper.

---

> ### Author Response · Authors · 2023-11-14
> **Author response to the reviewer (P2)**
>
> **3. multimodal robustness settings beyond visualization.**
>
> This is also a good point. First of all, joint multimodal pretraining in general helps improve the robustness of the modality-missing issue in the downstream tasks. As can be seen in our result tables reporting single-modal downstream scenarios (e.g. Table 1 right BarlowTwins-SAR v.s. BarlowTwins), pretraining with both modalities improves upon pretraining with only the downstream-available modality.
>
> Second, we do agree with your example that when one modality is severely perturbed, the encoded common features help but the encoded unique features from that modality can harm. However, we think such useless unique features provide less harm to the model compared to not decoupling. In a downstream case like this, we expect the model can efficiently ignore the unique (harmful) features but focus more on the common features as they are decoupled during pretraining and potentially easy to separate. To check that, we did a small experiment as shown below:
>
> | BigEarthNet-MM linear probing with 1% labels | BarlowTwins (no decoupling) | DeCUR (decoupling) |
> |:--------------------------------------------:|:---------------------------:|--------------------|
> |              before adding noise             |             78.7            | 79.4               |
> |        add noise to the optical images       |         75.6 (-3.1)         | 77.3 (-2.1)        |
>
> We add strong Gaussian noise to the optical images in the SAR-optical downstream dataset BigEarthNet-MM, and check how the model is affected. It can be seen that though both BarlowTwins and DeCUR's performance is decreased, the one with unique-representaton-decoupling (our DeCUR) is less affected.
>
>
> **4. Potential on other non-vision modalities.**
>
> Indeed we focus on visual modalities in this work, however we believe the method is not limited to vision and can be easily transferred to audio and texts. Since the learning process is fully conducted in the latent space, the format of the input modality doesn’t matter, as long as there is meaningful common/unique information. Take audio-text in a dialogue as one specific example, the common information can be the general semantic meaning of the topic, the unique information of audio can be the emotional contexts from tone, pitch, and modulation of voice, and the unique information of text can be specific names, historical or background information, and abstract concepts. In terms of training implementation, we just need to encode audio and texts into common latent space feature vectors and then everything is the same as in vision modalities. As mentioned in Point 2, extending to more modalities (vision+audio+language+etc.) is indeed a good follow-up.
>
> **Summary**
>
> Thanks again for the constructive comments! Hope the above responses can address your questions. We are also happy to answer any further questions. Looking forward to your feedback!

---

### Official Review · Reviewer_ZvJb · 2023-10-31

**Soundness:** 3 good
**Presentation:** 3 good
**Contribution:** 3 good
**Rating:** 8
**Confidence:** 4

**Summary:**

This paper addresses the limitation of existing multimodal self-supervised learning methods, which only focus on learning cross-modal common representations while overlooking the training of modality-unique representations. The proposed DeCUR method decouples the common and unique representations by distinguishing between inter-modal and intra-modal embeddings, allowing the integration of complementary information from different modalities. The effectiveness of the method is validated on three multimodal scenarios and two downstream tasks, along with an analysis of interpretability.

**Strengths:**

1. The method is simple yet effective, and the supplementary materials provide ample resources for reproducing the results.
2. The interpretability analysis aids in understanding the effectiveness of the proposed method.
3. The experiments are sufficient and comprehensive.

**Weaknesses:**

1.The results of training with 100% labels using Barlow Twins and VICReg are missing.

**Questions:**

NO

---

> ### Author Response · Authors · 2023-11-14
>
> Thanks for your positive feedback that strengthens our confidence! We believe this work offers valuable insights toward better understanding of multimodal representations.
>
> Regarding the results for 100% labels in Table 1 and 2, our thought was that the message is already clear from 1% labels and it is costly for end-to-end finetuning with 100% labels on all methods. However, we agree that full label results can make the conclusions more solid. We will update the full results soon (before the rebuttal deadline) in the updates!

---

### Author Response · Authors · 2023-11-18
**General updates for the revision**

Dear reviewers,

We would like to express our sincere gratitude for your valuable comments and suggestions. We have carefully considered and addressed each point raised in your reviews as shown in the separate responses. We're happy to answer any further questions.

In addition to the specific responses, we have finished an updated version of the manuscript with the following main revisions:
- clarify and highlight our technical contributions
- add important references
- add experimental results and comparisons
- add a discussion about limitations and future work
- reorganize the texts for space and correct a typo

We have highlighted the main revisions in blue color and the important clarifications in red color. You can also use the revision comparison tool in OpenReview to check the full revisions.

Thanks again for your constructive comments and we look forward to your feedback!

---

### Meta-Review · Area_Chair_maBL · 2023-12-17

**Metareview:**

The proposed DeCuR addressed the multi-modal self-supervised learning issue, which decouples the common and unique representations by distinguishing between inter-modal and intra-modal embeddings. Even though  Reviewer ZvJb gives positive, but quite short comments, other three reviewers all concerned about important problems and gave negative comments on: the method difference with Barlow Twins, the similarity with combination of CLIP and SimCLR, the applicability to other modalities, missing ablations. After the discussion stage and rebuttal stage, authors did not give satisfactory feedback. Thus ACs decide to reject it.

**Justification For Why Not Higher Score:**

The method contribution is quite limited.

**Justification For Why Not Lower Score:**

N/A

---

### Decision · Program_Chairs · 2024-01-16

Reject